# Probiotic Milk and Oat Beverages with Increased Protein Content: Survival of Probiotic Bacteria Under Simulated In Vitro Digestion Conditions

**DOI:** 10.3390/nu16213673

**Published:** 2024-10-29

**Authors:** Małgorzata Pawlos, Katarzyna Szajnar, Agata Znamirowska-Piotrowska

**Affiliations:** Department of Dairy Technology, Institute of Food Technology and Nutrition, College of Natural Sciences, University of Rzeszow, Cwiklinskiej 2D, 35-601 Rzeszow, Poland; kszajnar@ur.edu.pl (K.S.); aznamirowska@ur.edu.pl (A.Z.-P.)

**Keywords:** *Lactobacillus johnsonii*, *Lacticaseibacillus casei*, protein isolates, survival of probiotic bacteria, in vitro digestion

## Abstract

Background: The increasing prevalence of plant-based dietary preferences, driven by lactose intolerance, allergies, and adherence to vegan diets, has necessitated the exploration of alternative food matrices for probiotic delivery. Objectives: This study aimed to evaluate the effects of whey protein isolate, pea protein isolate, and soy protein isolate on the viability of *L. casei* and *L. johnsonii* during simulated in vitro gastrointestinal digestion. Furthermore, the study investigated the impact of two distinct matrices—cow’s milk and an oat-based beverage—on the survival of these probiotic strains. Fermented products were prepared using cow’s milk and an oat-based beverage as matrices, with simulated digestion performed following a seven-day storage period at 5 °C. The in vitro digestion model encompassed oral, gastric, and small intestinal phases, with probiotic viability assessed using the plate-deep method at each stage. Methods: Before digestion, *L. casei* exhibited higher populations than *L. johnsonii* in both matrices. Including 3% soy and pea protein, isolates promoted the growth of *L. casei* in both fermented milk and oat beverages. However, a marked reduction in probiotic viability was observed during the gastric phase, with *L. casei* counts decreasing by 6.4–7.8 log cfu g^−1^ in fermented milk and 3.1–4 log cfu g^−1^ in oat beverages, while *L. johnsonii* demonstrated similar reductions. Conclusion: These findings underscore the protective role of dairy components on probiotic viability, while the oat-based matrix exhibited a reduced capacity for sustaining probiotic populations throughout digestion. Future research should focus on optimizing plant-based matrices to enhance probiotic stability during gastrointestinal transit.

## 1. Introduction

The bacterial microflora of the human gastrointestinal tract plays a crucial role in protecting against antigens and pathogens introduced through food. The gut microbiota of an adult primarily consists of four groups of bacteria, belonging to the species *Bacteroidetes* (23%), *Firmicutes* (64%), *Proteobacteria* (8%), and *Actinobacteria* (3%) [1,2]. The most desirable bacterial species possess the ability to synthesize enzymes that facilitate the breakdown and absorption of nutrients, as well as those that create a beneficial environment, capable of defending against bacteriophages and mitigating acute immune responses [3,4]. Live, non-pathogenic microorganisms, including strains of lactic acid bacteria such as *Lactobacillus*, *Enterococcus*, *Propionibacterium*, *Lactococcus*, and *Streptococcus salivarius*, as well as *Bifidobacterium* and the yeast *Saccharomyces boulardii*, exert a beneficial health impact on the host by stimulating Th1 (Type 1 T helper cells) responses and reducing Th2 (Type 2 T helper cells) immune responses. Probiotics, as live microorganisms constituting the natural gut microflora, can adhere to the intestinal epithelium and colonize the gut, antagonizing the effects of typical gastrointestinal pathogens. At the same time, they do not exhibit toxic or pathogenic effects on the host organism.

Since food is digested in the stomach for 1–3 h, depending on the type of food content, it can be assumed that the incubation time for bacterial cells in a model gastric juice should be approximately 2 h. Charteris et al. [5] indicate that the pH value of this environment is crucial for the survival of bacteria in the upper gastrointestinal tract. The tolerance of strains to the enzymes and low pH of the stomach is strain-dependent [6]. Intestinal juice, conversely, is secreted at a rate of 0.7 litres per day, with a pH of about 8 and a mineral salt concentration of approximately 0.5%. The factors determining the survival of bacteria in the lower gastrointestinal tract are bile and the presence of enzymes, which primarily affect cell lysis and membrane permeability [7]. The proapoptotic properties of probiotic bacteria are enhanced by prebiotics. During the metabolism of prebiotics in the colon, short-chain fatty acids are produced, which stimulate the growth of probiotic bacteria such as *Lactobacillus* and *Bifidobacterium*. Inducing apoptosis in the gastrointestinal tract by probiotic bacteria and the enhancing effect of prebiotics prevent the further progression of inflammatory changes in the mucosa towards neoplastic changes. Synbiotics appear to be the future of medicine; however, their impact on apoptosis in the gastrointestinal tract requires further study [8]. Probiotics are usually added to food or are available as dietary supplements, and the presence of a food matrix or coating in the supplement positively affects the viability of bacterial cells in the gastrointestinal tract [9,10,11,12,13]. Kos et al. [10] demonstrated the beneficial effect of adding a food matrix during the simulation of the passage of *L. acidophilus* M92 through the gastrointestinal tract. They found that the use of a food medium inhibited protease activity.

Whey proteins were identified as the best protective factor for lactic acid fermentation bacteria. Therefore, this study aimed to investigate how adding whey protein isolate, pea protein isolate, and soy protein isolate would affect the survival of *L. casei* and *L. johnsonii* during simulated in vitro digestion. Additionally, this study aimed to investigate how applying different food matrices, such as cow’s milk and oat beverage, would affect the survival of probiotic strains. Therefore, oat drink was chosen as an alternative food matrix for the multiplication, growth, and survival of *L. casei* and *L. johnsonii*.

Moreover, this study addresses the needs of consumers who avoid cow’s milk due to allergies or intolerances to its components or those who follow diets that exclude animal products. Such consumers can substitute cow’s milk with plant-based beverages such as oat drink. Oat beverages represent an optimal alternative for individuals with lactose intolerance, milk protein allergies, or those adhering to a vegan diet. These plant-based drinks are characterized by their low fat content and are frequently enriched with essential nutrients, including vitamins D and B12, as well as calcium, to closely replicate the nutritional profile of cow’s milk. Additionally, oat beverages possess a subtle, naturally sweet flavor, making them highly versatile in various culinary applications, such as in coffee, breakfast cereals, smoothies, and baked goods. Oat drink contains fiber, particularly soluble fractions, with *β*-glucans being the most prominent [14,15]. *β*-glucans play a significant role in treating gastrointestinal diseases such as gastritis and diarrhea and mitigating the effects of peptic ulcer disease [16]. These components are credited with a crucial role in stimulating the immune system, which is especially important for maintaining health in elderly individuals [17]. Literature indicates that 200 mL of oat drink provides approximately 1 g of *β*-glucan.

The innovative focus of this research is the investigation of oat-based beverages as a viable matrix for probiotic delivery, specifically examining how protein enrichment influences the survival and viability of probiotic bacteria during digestion. While previous studies have concentrated mainly on dairy matrices, this research addresses the increasing consumer demand for plant-based alternatives, reflecting the growing trend toward sustainable and functional nutrition. Enriching oat beverages with protein isolates offers a critical innovation relevant to individuals seeking enhanced protein intake, such as athletes, active consumers, and vegans looking for plant-based protein sources. Additionally, the incorporation of probiotics into these beverages enhances their functional properties, supporting gut health and immune function. By investigating the interaction between various plant-derived protein sources and probiotics in oat-based matrices, this study contributes to developing health-promoting, nutritionally optimized products that meet modern consumer expectations for both functionality and sustainability.

## 2. Materials and Methods

### 2.1. Materials

Fermented milk was prepared using 2% fat milk Łaciate (SM Mlekpol, Grajewo, Poland). The oat beverage was derived from an oat drink Bio Owies (11% of oat; Auchan SAS OIA, Villeneuve d’Ascq, France). Soy protein isolate, pea protein isolate, and whey protein isolate “Biały Puch” were sourced from F.H.U. “KDJ” s.c. (Tarnów, Poland). The probiotic starter culture *Lactobacillus johnsonii* LJ Delvo^®^Pro was obtained from DSM (Delft, The Netherlands), and *Lacticaseibacillus casei* 431 was sourced from Chr. Hansen (Hoersholm, Denmark).

MRS agar and peptone water were acquired from Biocorp (Warsaw, Poland). Sodium hydroxide and phenolphthalein were supplied by Chempur (Piekary Śląskie, Poland).

The study utilized the following enzymes and reagents: heat-stable α-amylase (TDF-100 A, 24,975 U/mL; Sigma-Aldrich, St. Louis, MO, USA), porcine stomach mucin (type II; Sigma-Aldrich, St. Louis, MO, USA), porcine gastric mucosa pepsin (250 U/mg solid; Sigma-Aldrich, St. Louis, MO, USA), porcine bile extract (Sigma-Aldrich, St. Louis, MO, USA), porcine pancreas pancreatin (8×USP; Sigma-Aldrich, St. Louis, MO, USA), anhydrous di-sodium hydrogen phosphate, analytical grade ≥99.0% (Na_2_HPO_4_; 141.96 g/mol; Chempur, Piekary Śląskie, Poland), di-potassium hydrogen phosphate (K_2_HPO_4_; 174.18 g/mol; Chempur, Piekary Śląskie, Poland), analytical grade sodium chloride ≥99.9% (NaCl; 58.44 g/mol; Chempur, Piekary Śląskie, Poland), 12 mol hydrochloric acid (HCl, Chempur, Piekary Śląskie, Poland), and 1 mol sodium hydroxide (NaOH, Chempur, Piekary Śląskie, Poland). All reagents employed were of analytical grade.

### 2.2. Beverages Production

The milk and oat beverages were assigned into experimental groups, to which protein isolates were incorporated, as outlined in Table 1. The beverages containing isolates and the control samples were subjected to homogenization (60 °C, 20 MPa) and re-pasteurized at 85 °C for 10 min. Subsequently, the samples were cooled to 37 ± 1 °C and inoculated with a single starter culture of either *L. johnsonii* or *L. casei* (Table 1). Each batch underwent inoculation with a previously activated starter culture at 40 °C for 5 h, added to the milk and oat drink at 5% (*w*/*w*) based on the method described by Szajnar et al. [18]. The inoculated samples were stirred, poured into 100 mL plastic cups, fermented at 37 °C until reaching a pH of 4.60 ± 0.10, and subsequently cooled to 5 °C (Cooled Incubator ILW 115, POL-EKO Aparatura, Wodzisław Śląski, Poland). The experiment was conducted in triplicate. In vitro digestion and microbiological analysis of fermented beverages were carried out after seven days of refrigerated storage (5 °C).

### 2.3. In Vitro Digestion Process

The in vitro gastrointestinal digestion was conducted following the protocols of Buniowska et al. [19] and Kowalczyk et al. [20], with appropriate modifications. Simulated digestion of fermented beverages was initiated after a seven-day period of cold storage at 5 °C. The three-phase in vitro digestion model encompassed sequential oral, gastric, and small intestinal stages, detailing what is presented explicitly in Figure 1. The digestion simulation began with the oral phase. A 50 mL portion of each beverage sample was transferred to a 100 mL dark glass container and combined with 5 mL of salivary enzyme solution. The salivary solution was prepared by dissolving 2.38 g of Na_2_HPO_4_, 0.19 g of K_2_HPO_4_, 8 g of NaCl, 100 mg/L of mucin, and 150 mg/L *α*-amylase with an enzymatic activity of 200 U/L in 1 L of distilled water. The resulting mixture of beverage and saliva was then adjusted to a pH of 6.75 ± 0.20 using HCl (12 mol/L) or NaOH (1 mol/L) and incubated in a shaking water bath at 37 °C with gentle agitation at 90 rpm for 10 min.

Following the oral phase, the gastric phase was initiated by adding 13.08 mg of pepsin to the sample, and the pH was reduced to 2.0 ± 0.20 by adding HCl (12 mol/L). The mixture was then incubated in a shaking water bath for 2 h at 37 °C with constant agitation at 90 rpm to simulate the conditions within the stomach.

For the intestinal phase, the oral and gastric digestive contents were combined with 5 mL of pancreatin solution (4 g/L) and bile salts (25 g/L) at a pH adjusted to 7.00 ± 0.20 (using HCl 12 mol/L or NaOH 1 mol/L). The incubation continued for an additional 2 h under the same conditions (37 °C, 90 rpm), simulating the small intestine environment.

### 2.4. Microbiological Analysis and Survival Rate

The enumeration of viable probiotic strains in fermented beverages was conducted before and during each phase of simulated in vitro digestion (oral, gastric, and intestinal). For each analysis, 10 g of the sample were mixed with 90 mL of sterile 0.1% peptone water solution, and serial dilutions were subsequently prepared. The plate-deep method with MRS agar was used for inoculation, followed by anaerobic incubation in a vacuum desiccator at 37 °C for 72 h utilizing the GENbox anaer system (Biomerieux, Warsaw, Poland) and an incubator (Cooled Incubator ILW 115, POL-EKO Aparatura, Wodzisław Śląski, Poland). Probiotic colonies were enumerated using a TYPE J-3 colony counter (Chemland, Stargard Szczeciński, Poland), and the results were reported as log CFU g^−1^ [20]. The survival rate (%) was calculated by comparing the number of viable probiotic colonies in the intestinal contents to those in the undigested sample, according to the following Equation [20]:$$\mathrm{Survival}\;\mathrm{rate}\;(\%)=\frac{\mathrm{Viable}\;\mathrm{counts}\;\mathrm{of}\;\mathrm{probiotic}\;\mathrm{bacteria}\;\mathrm{in}\;\mathrm{digested}\;\mathrm{sample}\;\times 100}{\mathrm{Viable}\;\mathrm{counts}\;\mathrm{of}\;\mathrm{probiotic}\;\mathrm{bacteria}\;\mathrm{in}\;\mathrm{non}-\mathrm{digested}\;\mathrm{sample}}$$

### 2.5. Statistical Analysis

Statistical analysis was performed using Statistica 13.1 software (StatSoft, Tulsa, OK, USA) to determine the mean and standard deviation. Tukey’s test was employed to evaluate the significance of differences between means (*p* ≤ 0.05). Additionally, one-way and multifactor analyses of variance (ANOVA) and the correlation coefficient were conducted.

## 3. Results and Discussion

To maintain their therapeutic properties, probiotic bacteria must endure harsh conditions, such as the stomach’s acidic environment and the presence of bile in the digestive tract [21]. These challenges include exposure to low pH levels during the fermentation process of milk products and subsequent interactions with gastric acid, followed by contact with bactericidal bile acids, which pose significant survival barriers as they pass through the gastrointestinal system [22,23].

To assess the impact of protein isolate additions (whey, soy, pea) on the viability of two probiotic strains (*L. casei* and *L. johnsonii*) during gastrointestinal transit, the bacterial cell counts were determined in milk and oat beverage before digestion. The results detailing the cell counts of *L. casei* and *L. johnsonii* are presented in Table 2, Table 3, Table 4 and Table 5. In fermented milk, the pre-digestion cell count of *L. casei* was 0.22–0.49 log cfu g^−1^ higher than that of *L. johnsonii*. Similarly, in the oat beverage, the number of *L. casei* cells was 0.96–1.48 log cfu g^−1^ higher than *L. johnsonii* before digestion. It was found that in fermented milk, adding 1.5% and 3% soy protein isolate, as well as 3% pea protein isolate, positively stimulated the growth of *L. casei*. In contrast, in the oat beverage, adding 3% of the tested isolates had a beneficial effect on the number of *L. casei* cells.

The addition of 1.5% soy protein isolate and 3% soy and pea protein isolates was found to significantly enhance the growth of *L. casei* in fermented milk (Table 2). In fermented oat beverages, the highest *L. casei* cell counts prior to digestion were observed in the OLCW3, OLCS3, and OLCP3 samples, which contained the highest concentrations of protein isolates (Table 3). The cell count of *L. johnsonii* in fermented milk before digestion showed a significant reduction in population due to the addition of protein isolates in most groups, except for LJW1.5 (Table 4). However, in the oat beverage, pea protein isolate stimulated the growth of *L. johnsonii* at both 1.5% and 3% concentrations (Table 5). A multifactorial analysis of variance revealed that the type of milk/beverage and the bacterial strain, as well as their interactions, had a significant effect on the bacterial cell counts before digestion (Table 6).

The results presented in Table 2, Table 3, Table 4 and Table 5 indicate that by the end of the oral phase, no significant changes were observed in the cell counts of the strains in any of the test groups compared to the probiotic levels before digestion. After the gastric phase, there was a significant reduction in the probiotic populations across all milk and oat beverage groups (Table 7). In fermented milk with *L. casei*, the cell count decreased by 6.4–7.8 log cfu g^−1^, and in the oat beverage by 3.1–4 log cfu g^−1^, compared to the oral phase. Similarly, for *L. johnsonii*, the population in fermented milk decreased by 4.7–6.2 log cfu g^−1^, and in the oat beverage by 1.7–4 log cfu g^−1^ during the gastric phase compared to the oral phase. It is noteworthy that specific dairy components, such as whey proteins, have the capacity to protect lactic acid bacteria (LAB) and promote their survival. According to El Shafei et al. [24] the addition of whey proteins, either individually or in combination with starch, significantly enhanced the viability of *L. johnsonii* and its resistance to acidic conditions.

The addition of whey proteins (1.5% and 3%) favorably affected the survival of both *L. johnsonii* and *L. casei* during the gastric phase in milk and oat beverages. Increasing the soy protein isolate concentration from 1.5% to 3% resulted in a reduction in *L. johnsonii* populations in OLJS3 and LJS3. The opposite trend was observed for *L. casei*, where increasing the soy protein isolate dose led to an increase in the probiotic population in OLCS3 and LCS3 during the gastric phase. However, increasing the pea protein isolate to 3% significantly reduced the *L. casei* population in milk and oat beverages during the gastric phase. When analyzing the cell count of *L. johnsonii*, a higher probiotic count was observed in OLJP3 compared to OLJP1.5 in the gastric phase, whereas in milk, the opposite trend was observed for LJP1.5 and LJP3. The multifactorial analysis of variance indicates that all analyzed factors (bacterial strain, type of beverage, type of isolate, and isolate dose) significantly influenced the bacterial cell count during the gastric phase (Table 6). However, only the interactions between the bacterial strain and the type of beverage (*p* = 0.0000), as well as the type of beverage and the type of isolate (*p* = 0.0000), were found to be significant.

Probiotic microorganisms encounter significant gastrointestinal challenges as they pass through the human digestive tract [25]. One of these major challenges is acid stress, which arises as they move through the stomach, where the low pH is primarily due to hydrochloric acid in gastric juice [26]. Acid stress can substantially harm cellular structures, including the cell membrane, DNA, and proteins. As a result, acid resistance is a critical factor in selecting suitable probiotics [27,28]. Moreover, the lactic acid and other organic acids present in or produced by fermented dairy products used as probiotic carriers, due to the fermentation of lactose by LAB, lead to a decrease in milk pH, exacerbating acid stress [29]. Wu et al. [30] highlighted that alterations in membrane fluidity, distribution of fatty acids, and cell integrity are common strategies employed by *L. casei* to withstand severe acidification and mitigate the damaging impact of acid on the cell membrane. Furthermore, Hernandes-Hernandez et al. [31] and Nazzaro et al. [32] suggested that the choice of carbohydrate used as a carbon source during the cultivation of lactobacilli significantly affected the acid resistance of probiotic *Lactobacillus* strains under gastrointestinal conditions.

In this study, both strains (*L. johnsonii* and *L. casei*), originating from fermented milk, demonstrated better survival in simulated gastric juice with bile (Table 2 and Table 4). A significant increase in the population of *L. johnsonii* (from 1.56 log cfu g^−1^ to 3.5 log cfu g^−1^) and *L. casei* (from 1.57 log cfu g^−1^ to 3.34 log cfu g^−1^) was observed compared to their population size in the stomach stage. In the intestinal stage, pea protein isolate had the most favorable effect, while adding 3% soy protein isolate to milk significantly increased the *L. casei* population. Specifically, the 3% soy protein isolate addition to milk significantly increased the number of *L. casei* cells in the intestinal phase compared to the gastric phase.

This study’s significant finding is that milk is a superior carrier for probiotics (*L. johnsonii* and *L. casei*) compared to oat-based beverages. Analysis of both strains in the oat-based beverage during the intestinal phase showed a population reduction compared to the gastric phase (Table 3 and Table 5), whereas all groups of fermented milk showed population growth. The oat beverage containing *L. johnsonii* in the intestinal phase showed a lower cell count, ranging from 0.45 log cfu g^−1^ to 1.63 log cfu g^−1^, compared to the gastric phase. Similarly, in the intestinal phase, *L. casei* in the oat-based beverage exhibited a lower population, ranging from 0.24 log cfu g^−1^ to 1.88 log cfu g^−1^. In contrast, in the milk samples during the intestinal phase, >6.3 log cfu g^−1^ of *L. casei* cells and >7.2 log cfu g^−1^ of *L. johnsonii* cells were detected. In oat-based beverages, the counts were >4.9 log cfu g^−1^ and >5.0 log cfu g^−1^, respectively.

Multifactorial ANOVA analysis indicated that factors such as the type of beverage/milk (*p* = 0.0006), type of isolate (*p* = 0.0015), bacterial strain (*p* = 0.0218), and the interaction between the bacterial strain and type of isolate (*p* = 0.0241) significantly affected the probiotic cell count in the intestinal phase (Table 6). As reported by many authors [33,34], the survivability of probiotic strains gradually decreases during passage through the stomach and small intestine in vitro.

Our study evaluated survivability as a percentage by comparing the number of live cells in the intestine to the number of cells before digestion. An analysis of the survivability of *L. casei* from fermented milk revealed that the addition of 3% whey protein isolate increased it by 1.5%, while the addition of 3% soy protein isolate raised it by as much as 10.78% compared to the control group. In this case, adding only 1.5% protein isolate to the milk reduced the cell survivability of the strain. The addition of pea protein isolate also had a negative effect on the survivability of *L. casei* (Figure 2).

The survivability of *L. johnsonii* in milk samples was highly favorable. In most cases (except for LJP1.5), the applied doses of isolates enhanced the survivability of this probiotic strain compared to the control trial. However, changing the matrix from milk to an oat-based beverage had an adverse effect on the survivability of the *L. johnsonii* strain. Similarly, the use of protein isolate enrichment did not improve survivability.

Notably, the survivability of *L. casei* in the oat-based beverage was lower than in milk for the groups OLC, OLCW1.5, OLCW3, OLCS1.5, and OLCS3. However, adding pea protein isolate at 1.5% and 3% levels resulted in lower *L. casei* survivability in the oat beverages compared to their milk counterparts. Furthermore, only the oat beverage with 3% pea protein (OLCP3) demonstrated higher *L. casei* survivability than the control group (OLC).

Rzepkowska et al. [34] and Zaręba [35] reached similar conclusions, stating that the buffering properties of milk proteins positively affect the survival of probiotic *Lactobacillus* cultures. This is also corroborated by our observations regarding the change in matrix from milk to an oat-based beverage.

Research by Marttinen et al. [36] on enhancing the bioaccessibility of plant protein using probiotics demonstrated that probiotic counts were generally higher in soy and pea protein digests compared to whey protein digests. This was likely due to the lower solubility of soy and pea proteins in digestive juices compared to whey protein, thus providing better protection for bacteria against harsh digestive conditions, such as low pH. However, our findings do not confirm these results and indicate that the use of different probiotic strains and carriers alters the conditions, diminishing or eliminating the protective role of pea and soy proteins.

Probiotic bacteria themselves are capable of utilizing protein by breaking it down into smaller peptides and releasing free amino acids into their environment [37]. Peptidases produced by bacteria can either be secreted into the extracellular environment or released into the gastrointestinal environment when the bacteria undergo lysis [38,39]. These actions depend on the probiotic’s proteolytic mechanism and vary by species and strain. In lactic acid bacteria (LAB), the proteolytic system involves three key components: a cell-envelope-bound proteinase that initiates the hydrolysis of extracellular protein, specific peptide and amino acid transport systems, and various intracellular peptidases [37,40]. The genomes of LAB, including *Lactobacillus* species and related genera, as well as *Lactococcus* species, generally encode a large number and variety of proteases, peptidases, amino acid permeases, and transport systems [37,40].

Kos et al. [10] demonstrated the beneficial effect of adding a food matrix during the simulation of *Lactobacillus acidophilus* M92 passage through the gastrointestinal tract, showing that the use of a food medium inhibited protease activity. Whey proteins were found to be the best protective agent for lactic acid bacteria during fermentation.

### Study Strengths and Limitations

This study explores the potential of oat-based beverages as an innovative matrix for probiotic delivery, expanding the scope beyond the traditionally studied dairy products. By incorporating protein isolates such as whey, soy, and pea, the research responds to the increasing demand for plant-based, high-protein products, particularly among those seeking sustainable alternatives. The investigation into the survival of *L. casei* and *L. johnsonii* under simulated gastrointestinal conditions provides valuable insights into how different beverage matrices and protein isolates influence probiotic viability during digestion. The results suggest that while plant-based matrices offer promising alternatives, their ability to protect probiotics may differ significantly from dairy-based systems, mainly due to the buffering effects of components like whey protein. Additionally, the study reveals variability in the interaction between probiotics and specific protein isolates, indicating that certain combinations may not consistently enhance bacterial survival. Given that the research focuses on two probiotic strains, further studies are required to assess the broader applicability of these findings, including the use of other probiotic strains and plant-based matrices to optimize the formulation of functional beverages.

## 4. Conclusions

The research on fermented plant-based beverages faces significant challenges in improving processing techniques that ensure product safety while preserving nutritional and sensory qualities. The popularity of products with increased protein content continues to rise, alongside the introduction of new milk substitutes and various flavors. Food manufacturers can provide consumers with healthier options for high-protein products containing probiotics, which offer therapeutic effects similar to those of probiotic strains delivered through medications and dietary supplements.

The conducted studies indicate that fermented milk serves as a better carrier for *L. johnsonii* and *L. casei* than oat beverages. The results showed that in the small intestine phase, the fermented milk groups exhibited probiotic content ranging from 6.3 log cfu g^−1^ to 8.5 log cfu g^−1^. In contrast, the probiotic cell count in the oat beverage ranged from 4.9 log cfu g^−1^ to 6.7 log cfu g^−1^, indicating a lower potential of this beverage for delivering probiotics to enhance gut homeostasis. Unfortunately, increasing the protein content by adding isolates to the oat beverage, in most cases, contributed to a reduction in the survival of *L. johnsonii* and *L. casei* under simulated digestion conditions. The conducted studies suggest that fermented plant-based beverages may contain various compounds with distinct biological properties that can influence probiotic survival. However, further in vitro research is needed to assess the survival of other probiotic strains in high-protein beverages. Such an evaluation will be valuable in the development of new and innovative products with enhanced health benefits tailored to the needs of the gut ecosystem.

## Figures and Tables

**Figure 1 nutrients-16-03673-f001:**
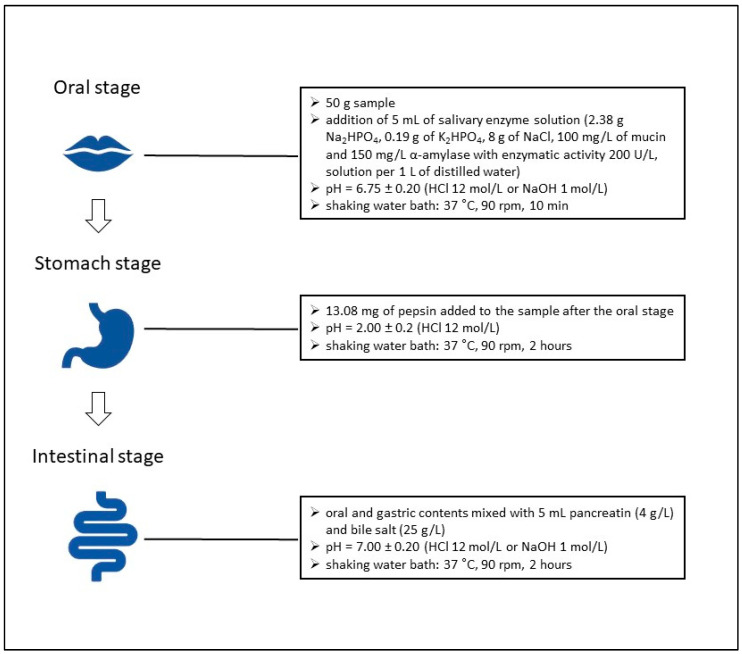
Procedure of in vitro digestion process.

**Figure 2 nutrients-16-03673-f002:**
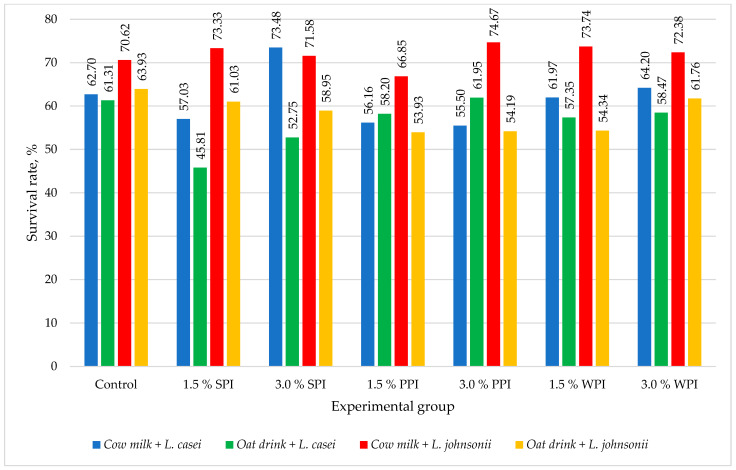
The survival rate of *L. johnsonii* and *L. casei* in fermented beverages depending on the isolate additive applied. WPI—whey protein isolate; SPI—soy protein isolate; PPI—pea protein isolate.

**Table 1 nutrients-16-03673-t001:** Experimental groups of fermented beverages.

Matrix	Probiotic Strain	Experimental Groups						
		Control	1.5% WPI	3.0% WPI	1.5% SPI	3.0% SPI	1.5% PPI	3.0% PPI
Cow milk	*L. casei* 431	MLC	MLCW1.5	MLCW3	MLCS1.5	MLCS3	MLCP1.5	MLCP3
	*L. johnsonii* LJ Delvo^®^Pro	MLJ	MLJW1.5	MLJW3	MLJS1.5	MLJS3	MLJP1.5	MLJP3
Oat drink	*L. casei* 431	OLC	OLCW1.5	OLCW3	OLCS1.5	OLCS3	OLCP1.5	OLCP3
	*L. johnsonii* LJ Delvo^®^Pro	OLJ	OLJW1.5	OLJW3	OLJS1.5	OLJS3	OLJP1.5	OLJP3

WPI—whey protein isolate; SPI—soy protein isolate; PPI—pea protein isolate.

**Table 2 nutrients-16-03673-t002:** *Lacticaseibacillus casei* cell count (log cfu g^−1^) in fermented milk depending on digestion stage.

Stage	Experimental Group						
	MLC	MLCW1.5	MLCW3	MLCS1.5	MLCS3	MLCP1.5	MLCP3
Before digestion	11.37 ^aA^ ± 0.22	11.36 ^aA^ ± 0.28	11.26 ^aA^ ± 0.31	11.59 ^bA^ ± 0.21	11.58 ^bA^ ± 0.20	11.27 ^aA^ ± 0.35	11.53 ^bA^ ± 0.25
Oral	11.30 ^aA^ ± 0.27	11.35 ^aA^ ± 0.51	11.62 ^abA^ ± 0.71	11.50 ^abA^ ± 0.53	11.69 ^bA^ ± 0.24	11.21 ^aA^ ± 0.42	11.47 ^abA^ ± 0.67
Stomach	4.53 ^bC^ ± 0.15	4.79 ^bC^ ± 0.38	5.17 ^cC^ ± 0.49	3.66 ^aC^ ± 0.17	5.17 ^cC^ ± 0.19	4.76 ^bC^ ± 0.33	3.88 ^aC^ ± 0.24
Small intenstine	7.13 ^cB^ ± 0.41	7.04 ^cB^ ± 0.37	7.23 ^cB^ ± 0.20	6.61 ^bB^ ± 0.22	8.51 ^dB^ ± 0.25	6.33 ^aB^ ± 0.27	6.40 ^aB^ ± 0.20

Mean ± standard deviation; ^a,b,c,d^—mean values denoted in rows by different letters differ statistically significantly at *p* ≤ 0.05; ^A,B,C^—mean values denoted in columns by different letters differ statistically significantly at *p* ≤ 0.05; MLC—control milk, MLCW1.5—milk with 1.5% WPI, MLCW3—milk with 3% WPI, MLCS1.5—milk with 1.5% SPI, MLCS3—milk with 3% SPI, MLCP1.5—milk with 1.5% PPI, MLCP3—milk with 3% PPI; *n* = 15.

**Table 3 nutrients-16-03673-t003:** *Lacticaseibacillus casei* cell count (log cfu g^−1^) in fermented oat beverage depending on digestion stage.

Stage	Experimental Group						
	OLC	OLCW1.5	OLCW3	OLCS1.5	OLCS3	OLCP1.5	OLCP3
Before digestion	10.47 ^aA^ ± 0.47	10.60 ^aA^ ± 0.18	10.86 ^bA^ ± 0.16	10.74 ^abA^ ± 0.08	10.88 ^bA^ ± 0.10	10.60 ^aA^ ± 0.48	10.83 ^bA^ ± 0.26
Oral	10.87 ^aA^ ± 0.80	10.62 ^aA^ ± 1.30	10.71 ^aA^ ± 1.12	10.72 ^aA^ ± 0.45	10.77 ^aA^ ± 0.62	10.56 ^aA^ ± 0.60	10.75 ^aA^ ± 0.72
Stomach	6.82 ^bB^ ± 0.21	6.63 ^aB^ ± 0.11	6.73 ^bB^ ± 0.03	6.80 ^bB^ ± 0.12	7.44 ^cB^ ± 0.03	7.46 ^cB^ ± 0.11	6.95 ^bB^ ± 0.03
Small intenstine	6.42 ^bB^ ± 0.20	6.08 ^abC^ ± 0.26	6.35 ^bC^ ± 0.19	4.92 ^aC^ ± 0.36	5.74 ^aC^ ± 0.59	6.17 ^abC^ ± 0.26	6.71 ^cC^ ± 0.19

Mean ± standard deviation; ^a,b,c^—mean values denoted in rows by different letters differ statistically significantly at *p* ≤ 0.05; ^A,B,C^—mean values denoted in columns by different letters differ statistically significantly at *p* ≤ 0.05; OLC—control oat beverage, OLCW1.5—oat beverage with 1.5% WPI, OLCW3—oat beverage with 3% WPI, OLCS1.5—oat beverage with 1.5% SPI, OLCS3—oat beverage with 3% SPI, OLCP1.5—oat beverage with 1.5% PPI, OLCP3—oat beverage with 3% PPI; *n* = 15.

**Table 4 nutrients-16-03673-t004:** *Lactobacillus johnsonii* cell count (log cfu g^−1^) in fermented milk depending on digestion stage.

Stage	Experimental Group						
	MLJ	MLJW1.5	MLJW3	MLJS1.5	MLJS3	MLJP1.5	MLJP3
Before digestion	11.37 ^bA^ ± 0.22	11.12 ^bA^ ± 0.25	10.79 ^aA^ ± 0.19	10.80 ^aA^ ± 0.20	10.77 ^aA^ ± 0.83	10.77 ^aA^ ± 0.17	10.82 ^aA^ ± 0.22
Oral	11.36 ^bA^ ± 0.17	11.10 ^bA^ ± 0.17	10.71 ^aA^ ± 0.13	10.76 ^aA^ ± 0.35	10.72 ^aA^ ± 0.21	10.68 ^aA^ ± 0.16	10.86 ^aA^ ± 0.37
Stomach	5.12 ^bC^ ± 0.10	5.61 ^cC^ ± 0.38	5.93 ^cC^ ± 0.16	5.75 ^cC^ ± 0.10	4.64 ^aC^ ± 0.33	5.64 ^cC^ ± 0.20	4.57 ^aC^ ± 0.30
Small intestine	8.03 ^cB^ ± 0.27	8.20 ^cB^ ± 0.12	7.81 ^bB^ ± 0.37	7.92 ^bB^ ± 0.45	7.71 ^bB^ ± 0.29	7.20 ^aB^ ± 0.29	8.08 ^cB^ ± 0.21

Mean ± standard deviation; ^a,b,c^—mean values denoted in rows by different letters differ statistically significantly at *p* ≤ 0.05; ^A,B,C^—mean values denoted in columns by different letters differ statistically significantly at *p* ≤ 0.05; MLJ—control milk, MLJW1.5—milk with 1.5% WPI, MLJW3—milk with 3% WPI, MLJS1.5—milk with 1.5% SPI, MLJS3—milk with 3% SPI, MLJP1.5—milk with 1.5% PPI, MLJP3—milk with 3% PPI; *n* = 15.

**Table 5 nutrients-16-03673-t005:** *Lactobacillus johnsonii* cell count (log cfu g^−1^) in fermented oat beverage depending on digestion stage.

Stage	Experimental Group						
	OLJ	OLJW1.5	OLJW3	OLJS1.5	OLJS3	OLJP1.5	OLJP3
Before digestion	9.29 ^abA^ ± 0.25	9.20 ^aA^ ± 0.15	9.31 ^bA^ ± 0.11	9.24 ^aA^ ± 0.07	8.99 ^aA^ ± 0.22	9.92 ^cA^ ± 0.15	9.78 ^cA^ ± 0.17
Oral	9.65 ^cA^ ± 0.31	9.04 ^aA^ ± 0.11	9.13 ^abA^ ± 0.15	9.25 ^bA^ ± 0.10	8.94 ^aA^ ± 0.18	9.86 ^cA^ ± 0.12	9.79 ^cA^ ± 0.44
Stomach	6.75 ^cB^ ± 0.52	6.51 ^bcB^ ± 0.41	7.38 ^dB^ ± 0.22	6.09 ^bB^ ± 0.33	5.83 ^aB^ ± 0.12	5.82 ^aB^ ± 0.18	6.31 ^bB^ ± 0.21
Small intenstine	5.94 ^bB^ ± 0.20	5.00 ^aC^ ± 0.33	5.75 ^bC^ ± 0.13	5.64 ^bC^ ± 0.21	5.30 ^aC^ ± 0.17	5.35 ^aC^ ± 0.14	5.30 ^aC^ ± 0.16

Mean ± standard deviation; ^a,b,c,d^—mean values denoted in rows by different letters differ statistically significantly at *p* ≤ 0.05; ^A,B,C^—mean values denoted in columns by different letters differ statistically significantly at *p* ≤ 0.05; OLJ—control oat beverage, OLJW1.5—oat beverage with 1.5% WPI, OLJW3—oat beverage with 3% WPI, OLJS1.5—oat beverage with 1.5% SPI, OLJS3—oat beverage with 3% SPI, OLJP1.5—oat beverage with 1.5% PPI, OLJP3—oat beverage with 3% PPI; *n* = 15.

**Table 6 nutrients-16-03673-t006:** Analysis of variance (ANOVA): *p*-values determining the effect of bacterial strain, beverage type, isolates dose and isolates type on the bacterial cell counts before digestion and in the oral, stomach, and small intestine stages.

Stage	Bacterial Strain	Beverage Type	Isolate Dose	Isolate Type	Bacterial Strain * Beverage Type	Bacterial Strain * Isolate Type	Beverage Type * Isolate Type	Beverage Type * Isolate Dose	Bacterial Strain * Isolate Dose	Isolate Type * Isolate Dose
Before digestion	0.0000 **	0.0003 **	0.1312	0.4097	0.0001 *	0.0590	0.1183	0.0069 **	0.0211 *	0.2941
Oral	0.0000 **	0.0003 **	0.2859	0.5678	0.0000 **	0.3190	0.5954	0.2025	0.3156	0.2777
Stomach	0.0000 **	0.0000 **	0.0055 **	0.0000 **	0.0000 **	0.0783	0.0000 **	0.0204 *	0.6432	0.7842
Small intenstine	0.0218 *	0.0006 **	0.2822	0.0015 **	0.0964	0.0039 **	0.0241 *	0.2879	0.1489	0.5834

Bacterial strain * Beverage type = interaction; Bacterial strain * Isolate type = interaction; Beverage type * Isolate type = interaction; Beverage type * Isolate dose = interaction; Bacterial strain * Isolate dose = interaction; Isolate type * Isolate dose = interaction; *—significant effect at *p* < 0.05; **—significant effect at *p* < 0.01.

**Table 7 nutrients-16-03673-t007:** Correlation coefficient.

	Before Digestion	Oral Stage	Stomach Stage
Oral Stage	0.62 *		
Stomach stage	−0.54 *	−0.55 *	
Small intenstine stage	0.10	0.10	−0.15

*—significant effect at *p* < 0.05.

## Data Availability

The data used to support the findings of this study can be made available by the corresponding author upon request.

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
