# Peer review of "Probiotic Milk and Oat Beverages with Increased Protein Content: Survival of Probiotic Bacteria Under Simulated In Vitro Digestion Conditions"

_nutrients, 2024, doi:10.3390/nu16213673_

Round 1
Reviewer 1 Report
Comments and Suggestions for Authors
The authors present in their manuscript in which way the different vegetal proteins (whey protein, pea protein, soy protein) affect the survival of L. casei and L. johnsonii during simulated in vitro digestion, and in which way the cow's milk and oat beverage influence the survival of the two probiotic strains. Comparatively, cow milk and oat drink were chosen as an alternative food matrix for the multiplication, growth, and survival of L. casei and L. johnson
The results indicated that fermented milk is a better carrier for L. johnsonii and L. casei than oat beverages. The results showed that in the small intestine phase, the fermented milk groups exhibited probiotic content ranging from 6.3 log cfu g⁻¹to 8.5 log cfu g⁻¹. In contrast, the probiotic cell count in the oat beverage ranged from 4.9 log cfu g⁻¹ to 6.7 log cfu g⁻¹, indicating a lower potential of this beverage for delivering probiotics to enhance gut microbiome.
Increasing the protein content by adding protein isolates to the oat beverage, reduces the survival of L. johnsonii and L. casei under simulated digestion conditions, probably due to the presence of various compounds from plants that influence probiotic survival, and here can be the novelty of this article.
However, some minor corrections are need before publication as follows:
1) The information presented in Table 1 to Table 6 is hard to read. Authors must present these tables in landscape format or must supply the large tables;
2 )Instead of Scheme 1, the authors must write Figure 1
3) Instead of Figure 1 authors must write Figure 2 ( below each mentioned figure, and in the manuscript also).
Author Response
The authors sincerely thank you for your time, for conducting the review, and for valuable comments.
Comments 1: 1) The information presented in Table 1 to Table 6 is hard to read. Authors must present these tables in landscape format or must supply the large tables;
Response 1: The tables in the manuscript have been formatted. Additionally, the tables have been submitted separately during the manuscript submission process for ease of review and clarity in presenting the data.
Comments 2: 2) Instead of Scheme 1, the authors must write Figure 1
Response 2: "Scheme 1" has been changed to "Figure 1".
Comments 3: 3) Instead of Figure 1 authors must write Figure 2 ( below each mentioned figure, and in the manuscript also).
Response 3: Corrections have been made throughout the manuscript.
Reviewer 2 Report
Comments and Suggestions for Authors
I believe the manuscript submitted by Pawlos et al. can be a good adition to Nutrients journal after the following revisions:
The abstract is too long. According to the journal’s guidelines the word limit is 250. It should start with a background statement and only then the authors should mention the study’s aims. At the end, please state some directions for further investigations.
More information about oat beverages should be given in the introductory section. A better justification and novelty of the study shoud also be provided in this section.
In section 2, “Scheme” should be replaced by “Figure”. This figure should be explained in detail by the authors.
Line 185: You should explain and discuss in detail each table (2-5).
Before Conclusions section I would suggest you include a subsection inside Results and Discussion about the main study strengths and limitations.
Supplementary Materials are missing.
Author Response
The authors sincerely thank you for your time, for conducting the review, and for valuable comments.
Comments 1: The abstract is too long. According to the journal’s guidelines the word limit is 250. It should start with a background statement and only then the authors should mention the study’s aims. At the end, please state some directions for further investigations.
Response 1: The abstract has been shortened and revised in accordance with the suggestions.
Comments 2: More information about oat beverages should be given in the introductory section. A better justification and novelty of the study shoud also be provided in this section.
Response 2: The manuscript introduction has been revised according to the suggestions (L103-109, L116-128).
Comments 3: In section 2, “Scheme” should be replaced by “Figure”. This figure should be explained in detail by the authors.
Response 3: The correction from "Scheme 1" to "Figure 1" has been made. Additionally, the procedure for the simulated in vitro digestion process has been discussed in section "2.3. In Vitro Digestion Process." (L173-190).
Comments 4: Line 185: You should explain and discuss in detail each table (2-5).
Response 4: In the specified section of the manuscript, the results concerning the cell counts of probiotic strains before digestion were discussed, with reference to all relevant tables, as per the suggestion (L232-239).
Comments 5: Before Conclusions section I would suggest you include a subsection inside Results and Discussion about the main study strengths and limitations.
Response 5: In response to the suggestion, a subsection discussing the main strengths and limitations of the study has been included in the "Results and Discussion" section, as recommended (L404-419).
Comments 6: Supplementary Materials are missing.
Response 6: In the manuscript supplementary materials were not planned. The information about their inclusion was mistakenly not removed. The mistake has been corrected.